# The Green Synthesis of Silver Nanoparticles from *Avena fatua* Extract: Antifungal Activity against *Fusarium oxysporum* f.sp. *lycopersici*

**DOI:** 10.3390/pathogens12101247

**Published:** 2023-10-16

**Authors:** Ahmad Kaleem Qureshi, Umar Farooq, Qaiser Shakeel, Sajjad Ali, Sarfraz Ashiq, Sohail Shahzad, Muhammad Tariq, Mahmoud F. Seleiman, Aftab Jamal, Muhammad Farhan Saeed, Barbara Manachini

**Affiliations:** 1Department of Chemistry, University of Sahiwal, Sahiwal 57000, Pakistan; drsohail@uosahiwal.edu.pk; 2Department of Chemistry, The Islamia University of Bahawalpur, Bahawalpur 63100, Pakistan; umarfarooqe2@gmail.com (U.F.); muhammadsarfrazashiq@gmail.com (S.A.); 3Cholistan Institute of Desert Studies, The Islamia University of Bahawalpur, Bahawalpur 63100, Pakistan; qaiser.shakeel@iub.edu.pk; 4Department of Entomology, The Islamia University of Bahawalpur, Bahawalpur 63100, Pakistan; sajjad.ali@iub.edu.pk; 5Institute of Chemical Sciences, Bahauddin Zakariya University Multan, Multan 60800, Pakistan; mtnazir@yahoo.com; 6Department of Plant Production, College of Food and Agriculture Sciences, King Saud University, P.O. Box 2460, Riyadh 11451, Saudi Arabia; mseleiman@ksu.edu.sa; 7Department of Soil and Environmental Sciences, Faculty of Crop Production Sciences, The University of Agriculture, Peshawar 25130, Pakistan; aftabses98@gmail.com; 8Department of Environmental Sciences, COMSATS University Islamabad, Vehari Campus, Vehari 61100, Pakistan; farhansaeed@cuivehari.edu.pk; 9Department of Agricultural, Food and Forest Sciences, University of Palermo, 90128 Palermo, Italy

**Keywords:** antifungal activity, *Avena fatua*, *Fusarium oxysporum*, green chemistry, silver nanoparticles

## Abstract

Using plant extracts as eco-friendly reducing and stabilizing agents for the synthesis of nanoparticles has gained significant attention in recent years. The current study explores the green synthesis of silver nanoparticles (AgNPs) using the *Avena fatua* extract and evaluates their antifungal activity against *Fusarium oxysporum* f.sp. *lycopersici* (*Fol*), a fungal plant pathogen. A green and sustainable approach was adopted to synthesize silver nanoparticles before these nanoparticles were employed for anti-fungal activity. The primary indication that AgNPs had formed was performed using UV-vis spectroscopy, where a strong peak at 425 nm indicated the effective formation of these nanoparticles. The indication of important functional groups acting as reducing and stabilizing agents was conducted using the FTIR study. Additionally, morphological studies were executed via SEM and AFM, which assisted with more effectively analyzing AgNPs. Crystalline behavior and size were estimated using powder XRD, and it was found that AgNPs were highly crystalline, and their size ranged from 5 to 25 nm. Synthesized AgNPs exhibited significant antifungal activity against *Fol* at a concentration of 40 ppm. Furthermore, the inhibitory index confirmed a positive correlation between increasing AgNPs concentration and exposure duration. This study suggests that the combined phytochemical mycotoxic effect of the plant extract and the smaller size of synthesized AgNPs were responsible for the highest penetrating power to inhibit *Fol* growth. Moreover, this study highlights the potential of using plant extracts as reducing and capping agents for the green synthesis of AgNPs with antifungal properties. The study concludes that *A. fatua* extract can synthesize antifungal AgNPs as a sustainable approach with robust antifungal efficacy against *Fol*, underscoring their promising potential for integration into plant protection strategies.

## 1. Introduction

Agriculture is facing significant challenges due to climate change and the rapid population increase. Difficulties meeting food demands are also due to yield losses caused by numerous plant pathogens and pests. Among plant diseases, fungi are one of the major yield-limiting factors, accounting for more than 25% of plant diseases [1]. The *Fusarium oxysporum* (Hypocreales: Nectriaceae) species complex has numerous strains pervasively inhabiting soil. *F. oxysporum* is one of the most destructive fungal plant pathogens in the world [2,3] as it causes damage to a wide range of crops, including date palm, oil palm, bananas, vegetables, flowers, cotton, and plantation crops. *Fusarium* wilt is one of the most significant plant diseases caused by *F. oxysporum* [4,5]. In addition to wilt disease, certain strains of *F. oxysporum* are also responsible for other diseases, including foot rot or root rot, leading to significant yield losses in many crops across the world [3,6].

The drawbacks of current management strategies against the diseases caused by *F. oxysporum* necessitate the development of alternatives to manage it. Recently, nanobiotechnology has emerged as an eye-captivating technology with successful outcomes against several plant diseases [7,8,9]. It is crucial to examine the impact and efficacy of nanosized particles on microorganisms as well as their use in the synthesis of fungicides and pesticides in order to fully utilize the advantage of nanotechnology in plant disease protection and management [10,11]. The creation of antimicrobial materials for the control of pathogenic organisms that affect agricultural crops, people, and animals has progressively benefited from the application of nanotechnology [12].

About 1300 nanoparticles (NPs) are reported with a wide range of possible applications [13]. Researchers’ attention has been drawn to significant advancements in the production of nanomaterials, such as polymeric, carbon-based, and metallic NPs, for use in treating pathogen-caused plant illnesses [14]. Nanoparticles, with sizes ranging from 1 to 100 nm, are commonly synthesized using both top-down and bottom-up approaches. In the bottom-up approach, tiny atomic particles are assembled to form nanoparticles, while in the top-down approach, bulk materials are degraded to meet demands [15,16,17,18].

Silver nanoparticles (AgNPs) exhibit antimicrobial activity because of their special antibacterial, physiochemical, catalytic, optical, and nano-pharmaceutical capabilities and attract much attention [19,20,21,22,23]. AgNPs also exhibit inhibitory efficacy against a variety of fungi that cause plant diseases [24]. Reducing and capping agents for the synthesis of NPs, plants, and plant extracts are more beneficial, straightforward, and economical than other biological primary sources [25]. Because it is a simple, useful, safe, and ecologically friendly one-step process, the plant-mediated synthesis of NPs is recommended [26,27,28]. In addition to various AgNP combinations, such as those with biomaterials [29], to inhibit microbial contamination, AgNPs are a viable choice in nanomedicine [30,31]. Therefore, AgNPs can be created using plant extracts, resulting in enhanced properties.

Several plant extracts were used to manufacture AgNPs, for example, from *Rosmarinus officinalis* L., *Urtica dioica* L., *Vaccinium vitis-idaea* L. [32], *Rheum palmatum* L. [33], *Excoecaria agallocha* L. [34], *Eucalyptus globulus* Labill [35], *Phyllanthus niruri* L. [36], *Vitex negundo* L. [37], *Datura metel* L. [38], *Dalbergia sissoo* Roxb. [39], *Nelumbo nucifera* Gaertn. [40], *Macrotyloma uniflorum* (Lam.) Verdc. [41], *Allium cepa* L. [42], peach gum [43], *Oxystelma esculentum* [44], *Glaucium flavum* Crantz [45], and *Solanum tuberosum* L. [46]. The prepared nanoparticles possess different morphologies like spherical, face-centered cubic, cluster triangular, hexagonal, ellipsoidal, crystalline, random, and polydispersed spherical. The resulting AgNPs were considered for numerous applications in medical, environmental, consumer goods, personal care products, food, electronic components, and transportation industries, and many of them exhibited antibacterial, antiviral, and antifungal potentials [47,48,49,50,51,52]. AgNPs were also synthesized from alga *Parachlorella kessleri* (Fott and Novakova) extracts and demonstrated antibacterial activity [53,54]. Previous studies also proposed that the green synthesis of nanoparticles could mitigate the detrimental consequences associated with conventional nanoparticle synthesis methods, frequently used in laboratory and industrial settings; thus, the green synthesis of nanomaterials is emerging as an essential alternative tool [55,56,57,58].

This study investigates the potential of *Avena fatua* L. (wild oat) extract as a green and sustainable method to synthesize antifungal AgNPs. *A. fatua* is a major wild weed found in many grain crops [59,60,61]. Thus, it can easily be used and procured for its use as a reducing agent in the production of nanoparticles. Here, it was used to reduce AgNO_3_ for the synthesis of AgNPs. Moreover, an extensive review of the literature for *A. fatua* showed many important classes of organic compounds. These classes included terpenoids, phenolic acids, aromatic, short-chain aliphatic organic acids, and many other unsaturated compounds, which provide an electron to reduce and cap Ag^+^ to Ag^0^ for the synthesis of AgNPs. Malic, succinic, fumaric, azelaic, p-coumaric acids, artemisia triene, and transcarveol are the major compounds of *A. fatua*, which are identified and quantified via gas chromatography and mass spectrometry [62,63]. To the best of our knowledge, *A. fatua* has never been used for the synthesis of any type of nanomaterials, nor has it been tested for its antifungal potential before. This study represents the first exploration of *A. fatua*’s antimicrobial potential against *F. oxysporum* f.sp. *lycopersici* through the greener synthesis of AgNPs.

## 2. Materials and Methods

### 2.1. Chemicals and Materials

Highly calibrated analytical grade silver nitrate (AgNO_3_, 99.9%) and ethanol (99%) were purchased from Merck and used without any further purification. Double-distilled de-ionized water was used in all aqueous solutions. All glassware was cleaned with distilled water and rinsed many times with deionized water.

### 2.2. Preparation of Plant Extract

About 1 kg of the *A. fatua* plant was collected from the periphery of Bahawalpur, Pakistan. The plant was washed thoroughly with double distilled water three times and rinsed with deionized water to ensure the complete removal of dust, impurities, and contaminants. After washing, the plant was dried under complete shade and ground into a fine powder using a kitchen blender. In total, 15 g of powdered *A. fatua* was mixed with 300 mL of deionized water using a magnetic stirrer hot plate (78-1 ISO certified, Jiangsu Jinyi Instrument Technology Company Limited, Changzhou, China). The mixture was stirred for 6 h at a constant temperature of 60 °C. Afterward, it was heated for one additional hour at the same temperature. The resultant extract was then filtered using Whatman filter paper grade 01 (pore size 25 µm) and stored at 4 °C in the incubator for further experimental work.

### 2.3. Synthesis of Silver Nanoparticles by Avena fatua Aqueous Extract

For the reaction, a 1 × 10^−2^ M AgNO_3_ solution in 100 mL of deionized water was prepared. In the AgNO_3_ solution, 100 mL of the *A. fatua* extract was added dropwise with the help of a burette. The reaction mixture was stirred continuously for 3 h at room temperature by a magnetic stirrer until a dark brownish color appeared, which physically indicated the formation of AgNPs. The reaction mixture was kept in complete darkness overnight for a complete reduction in Ag^+^ to Ag^0^. UV spectroscopic analysis was performed to confirm the AgNP synthesis. The concentration of AgNO_3_ was varied, while the concentration of the plant extract was kept constant. The process of AgNP synthesis using the *A. fatua* aqueous extract is illustrated in Figure 1.

### 2.4. AgNPs Characterization Techniques

The UV-spectrophotometer (Epoch, BioTek Instruments, Winooski, VT, USA) measured UV-visible absorption spectra graphs. Fourier transform infrared (FTIR) spectroscopic analysis was conducted using a TENSOR 27 instrument from Bruker for functional group identification. The crystalline metallic silver nature was observed using an X-ray powder diffraction (XRD) technique (XRD, Rigaku, Ultima IV, X-ray Diffractometer System), covering a 2-theta degree range of 10–80 degrees. Field emission scanning electron microscopy (SEM) was employed to investigate the morphology of synthesized AgNPs (silver nanoparticles) using a scanning electron microscope (JSM 6360 LA, JEOL, Tokyo, Japan). Energy dispersive X-ray spectrometry (EDX) analysis was used to examine the metallic nature of the nanoparticles. Finally, atomic force microscopy (AFM) was utilized to obtain high-resolution images of the topography of AgNPs.

### 2.5. Antifungal Bioassay

The *F. oxysporum* f.sp. *lycopersici* (*Fol*), used for the antifungal activity, was graciously provided by the Fungal Molecular Biology Lab and Fungal Molecular Biology Laboratory Culture Collection (FMB-CC-UAF), University of Agriculture, Faisalabad, Pakistan. *Fol* was grown in Petri dishes containing a potato dextrose agar (PDA) (Sigma-Aldrich, Merck KGaA, Darmstadt, Germany) at 25 °C for 15 days. Antifungal activity was evaluated via the poisoned food technique [48,49]. According to standard procedure, sterilized PDA was used for antifungal activity. Sterilized Petri dishes (90 mm) were taken in a laminar flow chamber. The *A. fatua* extract, AgNO_3_, and AgNPs at concentrations of 5 ppm, 10 ppm, 20 ppm, and 40 ppm were poured into Petri dishes with the help of a pipette and gently shaken so that each concentration had a smooth spreading of media while plates containing only PDA media served as the control treatment. Three replications for each treatment were performed. Media were allowed to solidify under UV light in a contamination-free environment. After solidification, mycelial discs (5 mm diameter) from *Fol* fungal culture were aseptically placed in the center of the plates. The plates were wrapped and incubated under 25 °C for 5 days. The fungal colony diameter was measured after 48, 72, 96, and 120 h.

The percentage of inhibition in growth was calculated using the following formula:$$Mycelial\;growth\;inhibition\left(\%\right)=[(dc-dt)/dc]\times 100$$
where dc and dt represent the average diameter of the fungal colony in control and treated Petri dishes, respectively [64].

### 2.6. Statistical Analysis

All antifungal bioassay treatments were replicated three times with a control. Data for mycelium growth (mm ± S.E.) and inhibitory index (%) for *Fol* were subjected to factorial ANOVA using Statistix 8.1 software. The means were compared using Tukey’s HSD test at a 5% level of significance to evaluate the impact of treatments on *Fol* mycelium growth and inhibitory index.

## 3. Results

### 3.1. UV-Visible Spectroscopy

Figure 2 shows UV-visible spectra of the green synthesis of AgNPs using a constant 100 mL *A. fatua* aqueous extract concentration and varying volume fraction concentrations of 1 × 10^−2^ M AgNO_3_, ranging from 150 mL (A) to 100 mL (B). It is observed that a characteristic surface plasmon resonance (SPR) absorbance band appears between 400 nm and 450 nm.

### 3.2. Fourier Transform Infrared Spectroscopy (FTIR)

The formation of AgNPs from *A. fatua* aqueous extract was clearly indicated using the results of Fourier transform infrared (FTIR) spectra (Figure 3). FTIR results indicated characteristic peaks that provided information on different functional groups from *A. fatua*. These different characteristic peaks include 3628.09 cm^−1^ (phenolic), 3316.65 cm^−1^ (alcoholic group), 2919.56 and 2850.82 cm^−1^ (aldehyde), 2146.20 cm^−1^ (Alkyne), 2357.70 cm^−1^ (α, β-unsaturated ketone), 1740 cm^−1^ (carbonyl aldehyde and ketone), 1636.94 cm^−1^ (alkenes), and 1540.7 cm^−1^ (esters), 1394.57 cm^−1^ (acetal), and 1227.57 cm^−1^ (alky aryl ether). These functionalities reduced the silver cations to AgNPs.

### 3.3. X-ray Diffraction (XRD)

Figure 4 interprets the XRD spectra of AgNPs synthesized using the *A. fatua* aqueous extract. It can be perceived that four characteristic peaks appeared with Bragg’s 2θ values of 31.5°, 45.7°, 67.2°, and 76.5°). Figure 4 (B) has quite a similar peak intensity and Bragg’s angle to that of Figure 4 (A). These peaks correlate to the Ag (111), Ag (200), Ag (220), and Ag (311) specific crystallographic planes of the face-centered-cubic geometry of AgNPs.

### 3.4. Scanning Electron Microscopy (SEM)

Qualitative and quantitative microscale analysis of synthesized AgNPs was conducted using scanning electron microscopy (SEM) and energy dispersive X-ray spectrometry (EDX). The SEM image of synthesized AgNPs is shown in Figure 5, which illustrates the morphology, including the shape and size of the nanoparticles. Phytochemical constituents of *A. fatua* capped the AgNPs, as revealed by the round-shaped nanoparticles that were well dispersed throughout the picture (depicted as white dots in Figure 5A). Synthesized AgNPs have a spherical morphology with an average diameter ranging from 25 nm to 40 nm. The elemental identification EDX profile of phytocapped silver NPs exhibited a characteristic Ag metal peak at a 3 keV value, as shown in Figure 5B.

### 3.5. Atomic Force Microscopy (AFM)

Atomic force microscopy (AFM) provided high-resolution images for the topography of AgNPs. The 3D AFM images of well-dispersed AgNPs are shown in Figure 6, depicting the presence of well-capped silver nanoparticles. Moreover, it can be observed that nanoparticles are spherical in shape, and the average size of these nanoparticles was found to be 0.22 nm. This utterly small size could also come in handy during biological activity as small-sized nanoparticles may have an advantage in terms of penetrating the fungal cell membrane or wall.

### 3.6. Antifungal Activity of Green Synthesized AgNPs

The antifungal activity of nanoparticles against the fungus *F. oxysporum* f.sp. *lycopersici* (*Fol*) is shown in Figure 7. In *Fol*, an alteration induced by AgNPs in the size and morphology of fungal colonies was observed. In particular, a slightly cottony texture and a lower mycelium density were observed compared to the control. Different treatments at various concentrations determined the time-based mycelial growth inhibition (%) against *Fol* (Figure 8). Table 1 reports the results of statistical analysis on the impact of different concentrations of AgNO_3_, *A. fatua* extract, and green synthesized AgNPs on the fungal colony diameter at different times.

The results showed that, in the control treatment, the *Fol* diameter was 38.26 mm after 48 h. When using a concentration of 40 ppm, the application of AgNPs resulted in a significantly lower colony diameter of *Fol* (7 mm), while *A. fatua* extract and AgNO_3_ led to larger colony diameters (21.66 mm and 32.16 mm, respectively). *Fol* colony diameters were affected less by the lowest concentrations (5 ppm) of all the treatments except for AgNPs (30.67 mm). In the case of 10 and 20 ppm concentrations, AgNPs significantly affected the *Fol* colony growth with 28.16 mm and 19 mm colony diameters, respectively (Table 1). After 48 h, the results revealed that AgNPs had an immense effect on *Fol* mycelial growth inhibition (%) at all concentrations. Mycelial growth inhibition was calculated as 20.18, 26.68, 50.54, and 81.78% at 5, 10, 20, and 40 ppm concentrations of AgNPs (Figure 8). After 72 h, the impact of AgNPs on *Fol* was the highest compared to other treatments. The colony diameter was 11 mm when *Fol* was exposed to AgNPs at concentrations of 40 ppm. By contrast, introducing the *A. fatua* extract and AgNO_3_ into PDA plates at the same concentration resulted in colony diameters of 33.33 mm and 50.66 mm for *Fol*, respectively. By contrast, the *Fol* colony diameter was 67.62 mm in the control treatments (Table 1). After 72 h, the results revealed that AgNPs significantly affected *Fol* mycelial growth inhibition at all concentrations. Mycelial growth inhibition was calculated at 23.96, 30.92, 63.95, and 83.7% at 5, 10, 20, and 40 ppm concentrations of AgNPs. The *A. fatua* extract caused 45.62% mycelial growth inhibition at 40 ppm, while AgNO_3_ had the lowest effect by producing only 24.94% of mycelial growth inhibition at the same concentration (Figure 8).

It was observed that *Fol* colony growth was not significantly different after 96 and 120 h, as the colony diameters were almost similar (Table 1, Figure 8). After 120 h, the *Fol* colony diameters were recorded as 15.54 mm at 40 ppm, 36.04 mm at 20 ppm, 56.69 mm at 10 ppm, and 61.54 mm at 5 ppm when AgNPs were employed in PDA media. At the same time, *Fol* colony diameters were 46.70 and 63.36 mm, respectively, when the *A. fatua* extract and AgNO_3_ were added to PDA plates at a 40 ppm concentration. The *Fol* colony diameter was 77.36 mm in the control treatment after the same time interval (Table 1). After 96 and 120 h, the results indicated that AgNPs significantly inhibited *Fol* mycelial growth at all concentrations. Mycelial growth inhibitions were calculated as 20.47% at 5 ppm, 26.71% at 10 ppm, 53.44% at 20 ppm, and 79.95% at 40 ppm concentrations of AgNPs. *A. fatua* extract caused 46.64% mycelial growth inhibition at 40 ppm, while AgNO_3_ caused the lowest effect by producing only 18.09% mycelial growth inhibition at the same concentration (Figure 8). Results comprising all treatments after application showed that AgNPs had the highest mycelial growth inhibition against the fungus *Fol* at each concentration.

## 4. Discussion

Fungal plant pathogens can negatively affect the quality and yield of crops. Farmers mainly apply chemical fungicides due to their simple and straightforward use and effectiveness against fungal pathogens. However, their harmful effects on the environment and human health, along with the development of resistance, have drawn attention and led to the investigation of pathogen management strategies based on natural compounds and, recently, nanotechnologies, particularly, the green synthesis of nanoparticles. A novel strategy aims to recycle poisonous plant chemicals and use them as natural stabilizing capping agents for nanoparticles.

This study demonstrates that green synthesized AgNPs using A. fatua extract exhibit potent antimicrobial activity against *Fol* due to the presence of bioactive compounds of *A. fatua*. UV-visible spectroscopy confirms the formation of AgNPs (silver nanoparticles). The identification of green synthesized AgNPs is based on surface plasmon resonance (SPR) absorbance bands, which exhibit their highest intensity at 435 nm [65]. The SPR for AgNPs depends on concentration, time, and temperature. It was also observed that sample (A) yielded optimum results for the green synthesis of AgNPs, exhibiting the highest absorbance intensity in the SPR due to optimized reaction conditions. Conversely, a decrease in the SPR absorbance intensity indicated a reduction in the volume concentration of AgNO_3_. UV-visible spectra also revealed the uniform morphology of synthesized AgNPs [66,67].

FTIR spectroscopy was used as a powerful technique for identifying specific functional groups and their transformations resulting from polymerization. Collected data permitted the estimation of the polymer structure, according to Varga et al. [67]. The literature on *A. fatua* revealed a comprehensive investigation into its composition, highlighting the presence of essential oils, phenolics, short-chain aliphatic acids, and various secondary metabolites. Among these, terpenoids were the most abundantly identified secondary metabolites. In the *A. fatua* extract, the major terpenoids identified were monoterpene hydrocarbons, oxygenated monoterpenes, sesquiterpene hydrocarbons, and oxygenated sesquiterpenes. Additionally, other important bioactive constituents were previously reported in *A. fatua*, including 4-Hydroxy-2,4,5-trimethyl-2,5-cyclohexadien-1-one, Ferulic acid, p-coumaric acid, and primary and secondary amine [62,63]. p-Coumaric acid is a hydroxycinnamic acid; therefore, IR spectra of the monomer and polymer ought to indicate fluctuations in the absorbing capacity of the aromatic ring, phenolic OH (hydroxyl) groups, carboxyl groups, and methylene as a result of polymerization [68]. Organic compounds present in the *A. fatua* extract, such as alkene, amine, phenol, polyol, ketone, aldehyde, and carboxylic functional groups, demonstrate a strong tendency to reduce Ag+ ions and encapsulate AgNPs [62,63]. Furthermore, FTIR results indicate many characteristic peaks that give information on different functional groups. These different characteristic peaks include 3628.09 cm^−1^ (phenolic OH), 3273.64, 3316.65, and 3388.07 cm^−1^ for the state-bonded and free alcoholic group (O-H), 2919.56 and 2850.82 cm^−1^ which indicate aldehyde’s (O=C-H) bond, 2146.20 cm^−1^, which informs alkyne, 2357.70 cm^−1^ α, β-unsaturated ketone, 1740 cm^−1^ for the carbonyl aldehyde and ketone functional group, 1636.94 cm^−1^ alkenes, 1540.7 cm^−1^ NO_2_, 1394.57 esters, and acetal, 1227.57 for the alky aryl ether (C-O-C), 1370 cm^−1^, which indicates the presence of bonded C-CH_3_ alkane, and 1038.27 cm^−1^ and 1074.78, which donate the presence of a primary and secondary amine N-H bond. All these functionalities cause a reduction in silver cations to AgNPs [69].

Moreover, peaks recorded with the XRD spectra of synthesized AgNPs (Figure 4) confirmed their correlation to 111, 200, 220, and 311 planes of the face-centered-cubic geometry of AgNPs, which was compared with the standard diffraction card of JCPDS silver file No. 04-0783 [35,61,65]; and indicated that AgNPs have a crystalline nature. The average crystallite size ranged from 5 nm to 25 nm. These findings are consistent with the estimation of the crystal size of AgNPs and measured from the full width at half maximum (FWHM) data using Scherer’s equation D = 0.9λ/βCosθ where 0.9 is the shape factor, λ is the wavelength of a peak, β is FWHM in the radian and θ is the Bragg angle [70,71]). In fact, by using this equation, the crystallite size calculated from the highest intensity peak was 11.33 nm and 11.93 nm.

Qualitative and quantitative microscale analyses of synthesized AgNPs were also confirmed via scanning electron microscopy (SEM) and energy-dispersive X-ray spectrometry (EDX) [72]. The strong signal of the Ag atom confirmed the crystalline property of AgNPs. Other elemental signals, along with the Ag atom, arose from phytochemical constituents of the *A. fatua* extract, confirming similar results from different plant extracts [69].

In the present research, Atomic Force Microscopy (AFM) was performed to better describe and study AgNPs from *A. fatua*. AFM is the best technique to provide high-resolution images of the topography of solid surfaces from the micrometer to the nanometer [73]. Moreover, AFM also explains the grain size, area, volume, and boundaries in 2D and 3D imaging forms. AFM photographs depict the presence of well-capped silver nanoparticles [74].

In addition, the findings of this research highlight that biosynthesized AgNPs using *A. fatua* showed great potential activity against the fungal plant pathogen *Fol.* Several other studies demonstrated the efficacy of biosynthesized AgNPs using plants or their products, indicating that they are potential alternatives against phytopathogenic fungi, which also mitigate the detrimental impacts of chemical pesticides [75].

Moreover, the current study revealed that the varying concentrations of green AgNPs significantly inhibited the colony growth of *Fol* and were found to be more effective with increased concentrations. These findings are in accordance with earlier findings with different extracts or NPs [76,77]. These studies report that lower to higher concentrations of AgNPs significantly inhibit *Fol* mycelial growth and were found to be more efficient.

Researchers tested the antifungal efficacy of *Bergenia ciliata*-derived AgNPs against various fungi, and their findings showed that the NPs performed better than the *B. ciliata* extract alone. *Bipolaris sorokiniana*, a pathogen that causes spot blotches on wheat, was completely inhibited by biofabricated AgNPs at various doses. At the same time, 100% conidial germination was seen in the absence of these NPs [78,79]. In the current study, it has been found that *A. fatua* extracts increase the antifungal activity of AgNPs (Table 1).

AgNPs synthesized from *A. fatua* can be more effective against *Fol* with 80% mycelial growth inhibition compared to the *A. fatua* extract with 43% mycelial growth inhibition or AgNO_3_ alone with 18.09% mycelial growth inhibition. Correspondingly, green synthesized AgNPs using strawberry waste inhibited the mycelial growth of the fungus *Fol* by 40–50% [80]. Another antifungal bioassay showed that the fungal growth of *Alternaria alternata* was greatly inhibited by applying green synthesized AgNPs (100 ppm) using an aqueous leaf extract of *Trigonella foenumgraecum*. Antifungal activity was shown because of the fungal mycelium disruption at various points [80].

In this research, in addition to growth inhibition due to AgNPs, an alteration in the morphology (size, cottony texture, and mycelium density) of the fungal colonies was reported. The antifungal activity of green-synthesized AgNPs against *Fol* worked through various mechanisms. Nanoparticles penetrate and disrupt the cell membrane of fungal cells [81,82] and induce the synthesis of reactive oxygen species within fungal cells [83]. Moreover, AgNPs prevent enzymatic activity in the fungal cells [84] and bind to fungal DNA, leading to interruptions in DNA transcription and replication processes [85,86]. These AgNPs continuously release Ag^+^ ions into cellular components, resulting in mitochondrial dysfunction and reduced ATP synthesis and cellular respiration [87,88].

Furthermore, AgNPs initiate a mechanical cell death mechanism similar to apoptosis [89]. Biocomponents from the plant extracts contribute to the formation of AgNPs and antifungal activity [90,91]. AgNPs showed increased antifungal activity due to the presence of plant secondary chemicals [92]. The green AgNPs increased the material’s long-term durability by preventing particle aggregation in the presence of plant-capping agents [93]. The obtained results of in vitro experiments are in general agreement and promising when correlated with previously reported studies on the antimicrobial activities of silver nanoparticles [94,95,96,97,98,99,100,101,102,103,104]. According to previous reports, antifungal activity is generally enhanced when AgNPs are conjugated with plant-based compounds. It is also worth noting that the antifungal activity of AgNPs also depends on their shape and size, damaging the fungal cell membranes and causing fungal death. Different shapes and sizes of biosynthesized AgNPs also exhibit different antifungal activities based on different plant materials, which can reduce AgNO_3_ differently [22,105,106]. The application of green AgNPs triggered severe cellular deformation by interfering with unsaturated fatty acids, thereby raising the cell membranes’ permeability, leading to the loss of salts, proteins, water, and some intracellular substances, which affect the viability of fungal cells [107,108].

*A. fatua* contains numerous bioactive compounds like alkaloids, flavonoids, phenolics, saponins, and tannins [62,63,109]. These phytochemicals have been reported for intrinsic antifungal properties. The amalgamation of these *A. fatua* components with AgNPs generates a synergistic effect, enhancing their overall antimycotic efficacy [110]. They can interrupt fungal cell membranes, prevent the essential enzymes, and affect fungal metabolism, ultimately leading to fungal cell death [111]. However, it changed when AgNPs were mixed with fungistats; synergistic and antagonistic effects were also observed against *Bipolaris maydis* upon combining AgNPs with fludioxonil and propineb [112].

Some studies demonstrate the effectiveness of using NPs of various elements against *Fol,* such as AgNPs. This type of NP has been shown to be capable of inhibiting the growth of *Fol* colonies in a PDA medium over a period of 5 h under concentrations that are exponentially higher compared to the AgNPs evaluated in this research [113,114]. Furthermore, the antifungal activity of green-synthesized AgNPs is also dependent on time and concentration, as shown in Table 1 and Figure 8. The mycelium growth of *Fol* rose after 48, 72, 96 and 120 h. According to the data presented in Table 1, a consistent inhibition in *Fol* growth was evident after 96 h. Among all treatments other than the control, AgNO_3_ showed the least efficacy at each concentration. The inhibition of *Fol* mycelium growth could be ascribed to the synergistic impact of the possibly antifungal *A. fatua* aqueous extract, and the reduced sizes of green synthesized AgNPs possessed greater penetration capabilities.

## 5. Conclusions

The results of the present experiment demonstrate that AgNPs can be synthesized through the green approach using the *A. fatua* aqueous extract with a phyto-reduction in Ag^+^ to Ag^0^ at room temperature. The synthesized nanoparticles were characterized using UV-visible spectroscopy, FTIR, AFM, XRD, SEM, and EDX. The results showed that nanoparticles have a crystalline nature with sizes ranging from 5 nm to 25 nm. The synthesized AgNPs and *A. fatua* plant extracts were further tested against the fungus *Fol*. AgNPs were found to be more active against the fungus *Fol* than plant extracts from *A. fatua*. The highest antifungal activity of synthesized AgNPs was observed at a 40 ppm concentration with an inhibitory index of 79.95% after 96 h. It might be because of the combined effect of the phytochemically hazardous *A. fatua* aqueous extract and the smaller size of synthesized AgNPs that they have the highest penetrating power to inhibit the mycelium growth of *Fol*.

## Figures and Tables

**Figure 1 pathogens-12-01247-f001:**
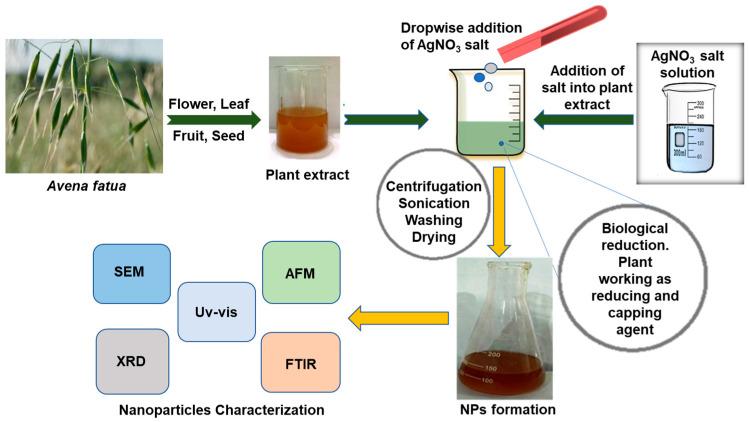
Schematic diagram of the synthesis of silver nanoparticles (AgNPs) and their characterization. AFM (atomic force microscopy), UV-vis (UV-spectrophotometry), SEM (scanning electron microscopy), XRD (X-ray powder diffraction), FTIR (Fourier transform infrared).

**Figure 2 pathogens-12-01247-f002:**
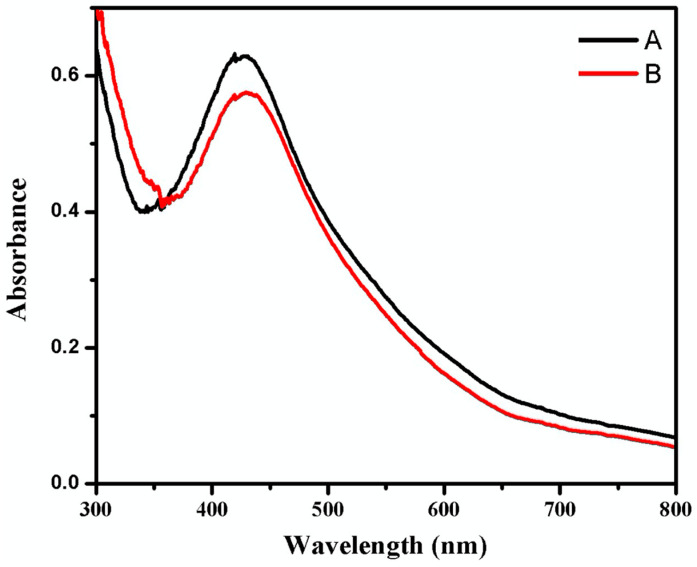
UV-visible spectra of AgNPs and its surface plasmon resonance (SPR) absorbance band excitation upon the interaction of *Avena fatua* aqueous extract with different concentrations at 150 mL (A) and 100 mL (B) of 1 × 10^−2^ M AgNO_3_.

**Figure 3 pathogens-12-01247-f003:**
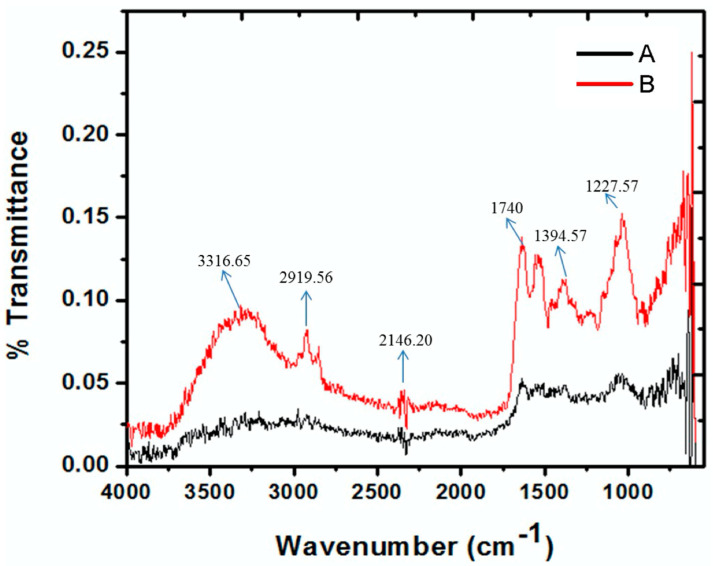
Fourier transform infrared (FTIR) spectra of *Avena fatua* aqueous extract (A) and AgNPs (B).

**Figure 4 pathogens-12-01247-f004:**
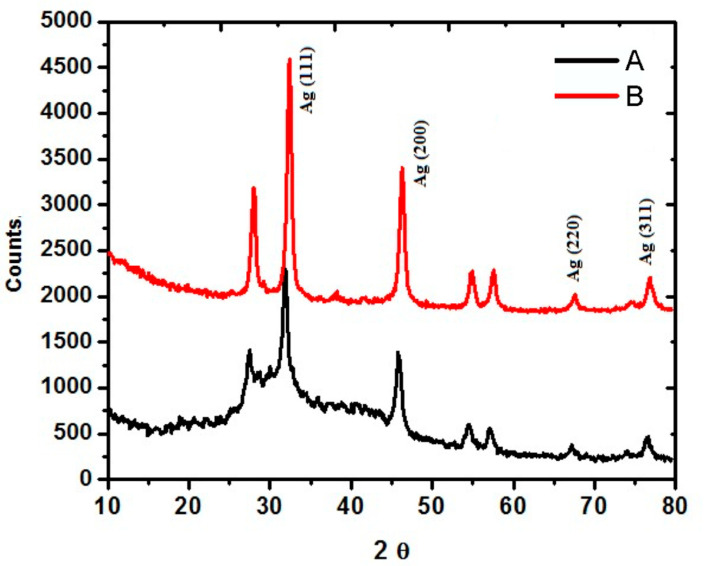
XRD spectra of synthesized AgNPs from *Avena fatua* aqueous extract (A) and AgNPs (B).

**Figure 5 pathogens-12-01247-f005:**
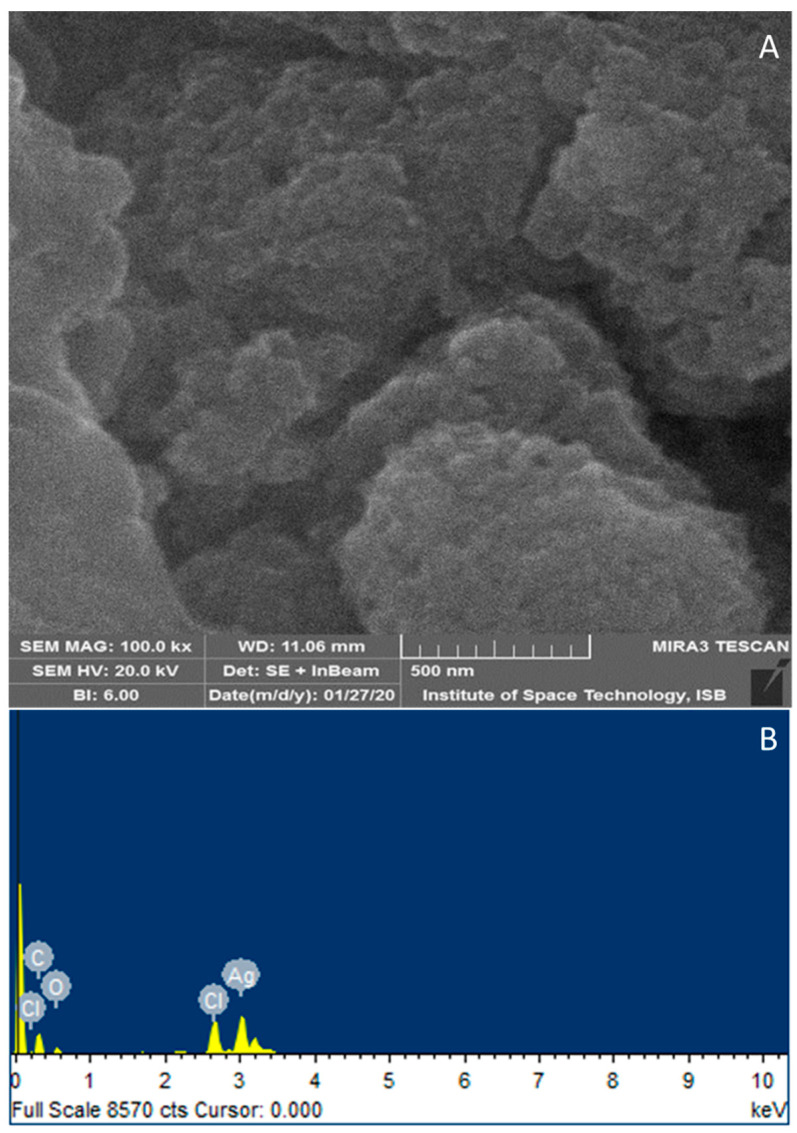
Scanning electron microscopy (SEM) (**A**) and energy dispersive X-ray spectrometry (EDX), (**B**) Image of synthesized AgNPs from *Avena fatua* aqueous extract.

**Figure 6 pathogens-12-01247-f006:**
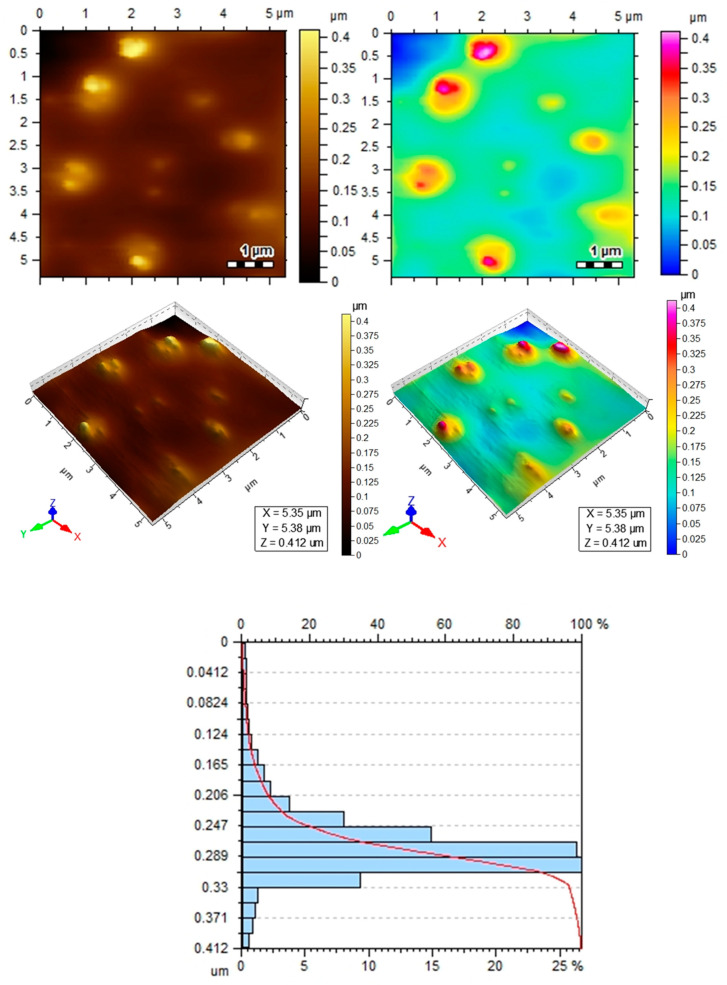
Atomic force microscopy (AFM) images obtained from prepared AgNPs using *Avena fatua* extract indicating the shape and size of AgNPs.

**Figure 7 pathogens-12-01247-f007:**
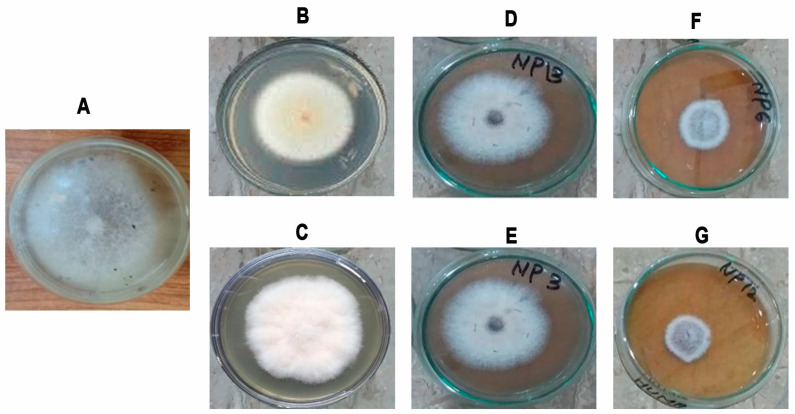
Inhibitory effect in *Fusarium oxysporum* f.sp. *lycopersici* mycelium growth. (**A**) Control without plant extract and AgNPs, (**B**) *A. fatua* plant extract, (**C**) 40 ppm AgNO_3_, (**D**–**G**) AgNPs concentrations: 5 ppm, 10 ppm, 20 ppm, and 40 ppm, respectively.

**Figure 8 pathogens-12-01247-f008:**
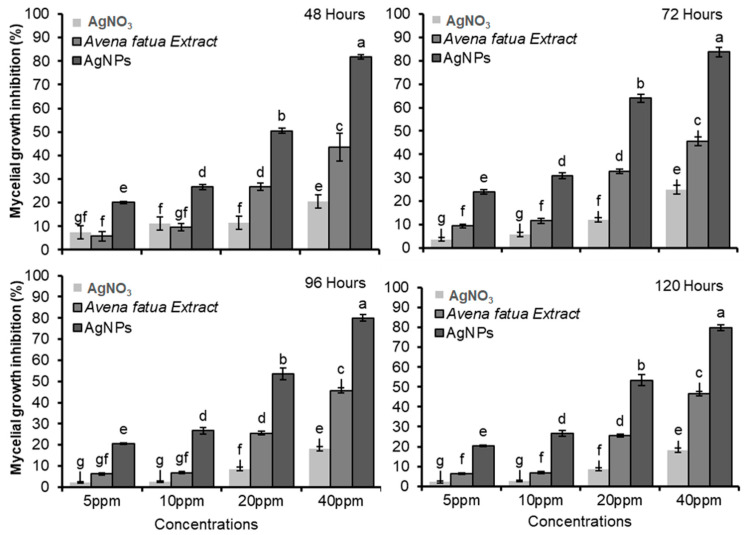
The mycelial growth inhibition (%) in *Fusarium oxysporum* f.sp. *lycopersici* in response to the application of different concentrations of AgNO_3_, the *Avena fatua* extract, and green-synthesized AgNPs after 48, 72, 96, and 120 h (df = 9, *p* = 0.0001). Significant differences are indicated with different letters.

**Table 1 pathogens-12-01247-t001:** Impact of different concentrations of AgNO_3_, the *Avena fatua* extract, and green-synthesized AgNPs on the fungal colony diameter (mm ± S.E.) of *Fusarium oxysporum* f.sp. *lycopersici* (*Fol*) after different time intervals.

**Concentrations**	**Treatments**	**Fungal Colony Diameter of *Fol* (mm)**			
		**48 h**	**72 h**	**96 h**	**120 h**
	Control	38.3 ± 0.23 ^k^	67.6 ± 0.35 ^cd^	77.3 ± 0.88 ^a^	77.3 ± 0.88 ^a^
**5 ppm**	AgNO_3_	35.5 ± 0.76 ^klm^	65.0 ± 0.57 ^de^	75.7 ± 0.33 ^ab^	75.7 ± 0.33 ^ab^
	*A. fatua* extract	36.2 ± 0.16 ^kl^	61.2 ± 0.60 ^efg^	72.5 ± 0.28 ^abc^	72.5 ± 0.28 ^abc^
	AgNPs	30.7 ± 0.60 ^mn^	51.3 ± 1.20 ^hi^	61.5 ± 1.04 ^efg^	61.5 ± 1.04 ^efg^
**10 ppm**	AgNO_3_	34.1 ± 0.59 ^klm^	63.7 ± 0.66 ^def^	75.3 ± 0.33 ^ab^	75.4 ± 0.33 ^ab^
	*A. fatua* extract	34.8 ± 0.44 ^klm^	59.7 ± 0.88 ^efg^	72.0 ± 1.15 ^abc^	72.1 ± 1.15 ^abc^
	AgNPs	28.2 ± 0.72 ^no^	46.7 ± 1.21 ^ij^	56.7 ± 1.76 ^gh^	56.7 ± 1.76 ^gh^
**20 ppm**	AgNO_3_	34.0 ± 0.57 ^klm^	59.3 ± 0.66 ^fg^	70.7 ± 0.67 ^bc^	70.7 ± 0.67 ^bc^
	*A. fatua* extract	28.2 ± 0.44 ^mn^	45.3 ± 1.20 ^j^	57.7 ± 2.02 ^g^	57.7 ± 2.02 ^g^
	AgNPs	19.0 ± 0.57 ^pq^	24.3 ± 0.88 ^op^	36.0 ± 1.52 ^klm^	36.0 ± 1.52 ^klm^
**40 ppm**	AgNO_3_	32.2 ± 0.60 ^lmn^	50.7 ± 1.20 ^ij^	63.3 ± 0.88 ^def^	63.4 ± 0.88 ^def^
	*A. fatua* extract	21.7 ± 0.33 ^p^	33.3 ± 1.45 ^k–n^	46.7 ± 1.20 ^ij^	46.7 ± 1.20 ^ij^
	AgNPs	7.0 ± 0.57 ^s^	11.0 ± 0.57 ^rs^	15.5 ± 1.32 ^qr^	15.5 ± 1.32 ^qr^

Different letters among rows and columns represent significant differences among the fungal colony diameter of mycelial growth from AgNO_3_, plant extract, and AgNPs, at the same concentration with a 5% significance level (df = 27, F = 5.72, *p* = 0.0000).

## Data Availability

The data presented in this study are available on request from the first corresponding author.

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
