# Peer review of "The Green Synthesis of Silver Nanoparticles from Avena fatua Extract: Antifungal Activity against Fusarium oxysporum f.sp. lycopersici"

_pathogens, 2023, doi:10.3390/pathogens12101247_

Round 1

Reviewer 1 Report

The authors evaluated the antifungal activity of green synthesized Ag NPs. The results are interesting. However, some explanations should be revised:

1- The manuscript contains several typographical as well as grammatical errors. Authors are suggested to get the manuscript proofread by a native English speaker.

2- It is an interesting topic of green synthesized metal nanoparticles, however, there are several similar publications, such as Scientific Reports 13 (1) (2023) 5987, Applied Physics A 127 (2021) 1-8, Nanomedicine Research Journal 5 (3) (2020) 265-275, Applied Nanoscience 13 (6) (2023) 4395-4405, Applied Physics A 129 (8) (2023) 564, and Journal of Drug Delivery Science and Technology 85 (2023) 104541. The authors should compare with such literature, and justify the difference or new approach/knowledge from the current study.

3- Some parameters need to be defined just after appearing in the text.

4- # Results: “From the SEM image, it can  be observed that the phytochemical constituents of A. fatua have capped the AgNPs, revealing their crystalline nature”. With which evidence was this conclusion reached? Explanation needed.

5- Determine A and B in Figures 3, 4, and 5 in the figure caption. Furthermore, explain the difference between A and B in these figures.

6- In XRD and EDX, more peaks are not identified or discussed in the text.

7- The author should propose the interactive relation between the bio-components and the  Ag NPs that give the studied properties and applications.  

8- The author should confirm the exact bio-components type and quantity that still attached on the Ag NPs surface, because such compounds could not be able to detect by SEM.

9- All the findings of the current work need to be compared and discussed with the results of other researchers finding instead of having a general comparison with other researchers' works. The authors should perform a comparison between the forecasting results. In your discussion section, please link your empirical results with a broader and deeper literature review.

10- What is the role of bio-components and Ag NPs in the antifungal activitiy of the nanoparticles? Explanation needed.

The manuscript contains several typographical as well as grammatical errors. Authors are suggested to get the manuscript proofread by a native English speaker.

Author Response

Dear Reviewer, #1,

Many thanks for pointing out the suggestions, providing valuable improve to better understanding our manuscript. We have completely revised the manuscript while removing the pointed flaws and considering the given directions. Moreover, the manuscript has been re-checked for linguistic/scientific quality. The said changes have been incorporated highlighted with yellow color.

Comments and Suggestions for Authors

The authors evaluated the antifungal activity of green-synthesized Ag NPs. The results are interesting. However, some explanations should be revised:

Response: We would like to extend our heartfelt gratitude to Reviewer for his invaluable contribution to the review process of our submitted manuscript and appreciation for our work. We have carefully addressed all the comments and suggestions that reviewer has raised. Please find below our point-by-point response to the reviewer’s comments/ suggestions and the necessary changes are highlighted with yellow color in the revised manuscript as well.

1- The manuscript contains several typographical as well as grammatical errors. Authors are suggested to get the manuscript to proofread by a native English speaker.

Response: Thank for your valuable suggestion. We are extremely sorry for these mistakes and be careful in future. We have now revised the whole manuscript and corrected the grammatical and punctuation mistakes. Please see this revised version of the manuscript.

2- It is an interesting topic of green synthesized metal nanoparticles, however, there are several similar publications, such as Scientific Reports 13 (1) (2023) 5987, Applied Physics A 127 (2021) 1-8, Nanomedicine Research Journal 5 (3) (2020) 265-275, Applied Nanoscience 13 (6) (2023) 4395-4405, Applied Physics A 129 (8) (2023) 564, and Journal of Drug Delivery Science and Technology 85 (2023) 104541. The authors should compare with such literature, and justify the difference or new approach/knowledge from the current study.

Response: Thank you for your thorough review and valuable guidance. We appreciate your feedback. We have carefully revised the introduction, incorporating additional relevant literature to strengthen the background and context of our study. Furthermore, we have included the hypothesis of the study to provide a clear research objective. We kindly request your evaluation of the updated manuscript, and we remain open to any further suggestions or improvements you may recommend. Please see line 92-96, and line 112-124.

3- Some parameters need to be defined just after appearing in the text.‎

Response: Thank you for bringing this to our attention. We apologize for any confusion caused by the lack of parameter definitions. In this revised version, we have taken care to define all parameters immediately after their first appearance in the text to ensure clarity and improve the understanding of our study.

4- # Results: “From the SEM image, it can be observed that the phytochemical constituents of A. fatua have capped the AgNPs, revealing their crystalline nature”. With which evidence was this conclusion reached? Explanation needed.

Response: Thank you for bringing this to our attention. Upon careful review, we acknowledge the oversight in our previous statement. You are correct that the crystalline nature of the nanoparticles should be examined using Powder X-ray Diffraction (XRD), not SEM. We apologize for the error and any confusion it may have caused.

In our revised manuscript, we have made the necessary correction to accurately describe the characterization techniques used. The crystalline nature of the nanoparticles was indeed analysed using Powder XRD, while SEM was utilized to investigate the morphology and size of the nanoparticles. Please see section 3.4 Scanning Electron Microscopy (SEM) in this revised version.

5- Determine, A and B in Figures 3, 4, and 5 in the figure caption. Furthermore, explain the difference between A and B in these figures.”

Response: Thank you for your kind suggestion. In Figure 3, A and B are two different concentrations dependent UV-vis spectra of the nanoparticle’s solution. Figure 3(A) denotes the solution with 150 mL of plant concentration, while Figure 3(B) illustrates the plant concentration of 100 mL in the solution. The increase in peak intensity hints at the increase in the size of the nanoparticles and their movement toward stability.

In Figure 4 (FTIR spectrum), Figure 4(A) indicates the FTIR spectra while Figure 4(B) exhibits the FTIR of AgNPs synthesized from it.

Furthermore, in Figure 5 (XRD spectra) the two different concentrations dependent synthesized nanoparticles were analyzed to determine if it caused any difference in their crystalline behavior or their size. The peak of the more concentrated solution (150 mL) (Figure 5 B) is more visible and clearer rather than A which can be an indication of the sample having a greater size and more crystallinic behavior.  

6- In XRD and EDX, more peaks are not identified or discussed in the text.

Response: Thank you for pointing it out. During the EDX analysis, we tried to obtain the characteristic peaks of more prevailing atoms rather than the trace elements, which would be much more difficult to mention and justify in the text. Those minor peaks depict the trace elements. While during the elucidation of XRD spectra, we matched the characteristic peaks with the JCPDS card of silver nanoparticles and thus mentioned only the distinctive peaks. Moreover, some of those minor peaks tend to appear from the untreated and unreacted silver nitrate salt during the reaction and sometimes XRD sample holder may also have some traces of previous sample remains which as well may include in the other samples being analyzed too.

7- The author should propose the interactive relation between the bio-components and the AgNPs that give the studied properties and applications. 

Response: Dear reviewer, we appreciate your valuable comment. It is well-known and reported in the literature that the bio-components present in the plant extract help in carrying, reducing, and stabilizing the entire reaction. So, during the greener route, we only use plant extract and basic metal precursor as plant extract (bio-components in it) plays a significant role in the reaction. In the other synthetic routes, reducing agents such as NaBH4, tri-sodium citrate, etc have been used which is comparatively toxic to the afore-mentioned method. After the adequate synthesis of the nanoparticles, a comparative study can be performed taking plant extract on one hand and then analyzing the nanoparticles on the other. For reference these articles can be considered:

  • Mallmann, E. J. J., Cunha, F. A., Castro, B. N., Maciel, A. M., Menezes, E. A., & Fechine, P. B. A. (2015). Antifungal activity of silver nanoparticles obtained by green synthesis. Revista do instituto de Medicina Tropical de sao Paulo57, 165-167.
  • Abkhoo, J., & Panjehkeh, N. (2017). Evaluation of antifungal activity of silver nanoparticles on Fusarium oxysporum. International Journal of Infection4(2).
  • Hawar, S. N., Al-Shmgani, H. S., Al-Kubaisi, Z. A., Sulaiman, G. M., Dewir, Y. H., & Rikisahedew, J. J. (2022). Green synthesis of silver nanoparticles from Alhagi graecorum leaf extract and evaluation of their cytotoxicity and antifungal activity. Journal of Nanomaterials2022, 1-8.
  • Anum, F., Jabeen, K., Javad, S., Iqbal, S., Tahir, A., Javed, Z., ... & Cho, W. C. (2021). Green synthesized silver nanoparticles as potent antifungal agent against Aspergillus terreus Thom. Journal of Nanomaterials2021, 1-10.

8- The author should confirm the exact bio-components type and quantity that are still attached on the AgNPs surface because such compounds could not be able to detect by SEM.

Response: Thank you for your kind opinion. It is correct that SEM is primarily used to visualize the morphology and surface characteristics of materials at a microscale level and may not provide detailed chemical information about the surface functionalization or attached bio-components. Thus, for that reason, we added FTIR analysis to the manuscript. It can be observed in Figure 4 that there are still important functional groups peaks in the spectrum which indicate the presence of those bio-components in the AgNPs. We are sorry that right now our institute has no access to avail the NMR and HPLC facility which is very important to evaluate the presence of those bio-components. But in the future, we can consider them in our incoming projects. We again appreciate your critical analysis as the addition of those analytical techniques would be much more advantageous. Present work can be compared with these articles:

  • Hawar, S. N., Al-Shmgani, H. S., Al-Kubaisi, Z. A., Sulaiman, G. M., Dewir, Y. H., & Rikisahedew, J. J. (2022). Green synthesis of silver nanoparticles from Alhagi graecorum leaf extract and evaluation of their cytotoxicity and antifungal activity. Journal of Nanomaterials2022, 1-8.
  • Ghojavand, S., Madani, M., & Karimi, J. (2020). Green synthesis, characterization and antifungal activity of silver nanoparticles using stems and flowers of felty germander. Journal of Inorganic and Organometallic Polymers and Materials30, 2987-2997.
  • Jebril, S., Jenana, R. K. B., & Dridi, C. (2020). Green synthesis of silver nanoparticles using Melia azedarach leaf extract and their antifungal activities: In vitro and in vivo. Materials Chemistry and Physics248, 122898.

9- All the findings of the current work need to be compared and discussed with the results of other researchers finding instead of having a general comparison with other researchers' works. The authors should perform a comparison between the forecasting results. In your discussion section, please link your empirical results with a broader and deeper literature review.

Response: Thanks for suggestion. Discussion part has been revised. In the discussion section broader and deeper literature review has been. Results are also compared with the other researchers works. Hopefully revision will satisfy your requirements.

10- What is the role of bio-components and AgNPs in the antifungal activity of the nanoparticles? Explanation needed.

Response: The bio-components from A. fatua contribute to the reduction and stabilization of AgNPs during the green synthesis process. These AgNPs, in turn, possess inherent antifungal properties, and their combination with bio-components may lead to a synergistic effect, enhancing the overall antifungal activity of the nanoparticles. The interplay between bio-components and AgNPs contributes to the efficacy of the green-synthesized nanoparticles against the targeted fungus, F. oxysporum, making them a promising candidate for eco-friendly and sustainable antifungal applications. Furthermore, the detail has also been included in the revised manuscript.

Reviewer 2 Report

The authors reported the biogenic synthesis of silveer nanoparticles and evaluated their antifungal activity against Fusarium oxysporum.

The work is well written, the methodology is correct and also the results and discussion. However, the most important aspect to consider this paper acceptable for publication is the novelty. Many works have reported the biogenic synthesis of metals nanoparticles and their antifungal properties it is no a novelty for the readers. Then the aunthors need to highlight the novelty of this manuscript or add new and more relevant information.

on the other hand, why the authors used AgNO3 as control? this material is not frecuently used as antifungal. Why the authors not used a common antifungal?

in Table 1, please only 1 decimal after point.

the english is good, only minor revisions

Author Response

Dear reviewer, 

thank you very much for your time and effort in improving our document. We accept all your useful suggestions. Please find below in detail our changes. 

 Comments and Suggestions for Authors

Considering my expertise (General microbiology), the current revision will cover only the antifungal tests. The main goal of the paper was to investigate the potential of Avena fatua as a green source of AgNPs with antifungal activity. However, the idea of using plants for the synthesis of silver nanoparticles with antimicrobial properties is not new and from this perspective, the main drawback of the manuscript is the lack of novelty. In addition, in this form, the manuscript is unacceptable for publication as it does not comply with the minimum scientific standards of writing a paper.

Response: Thank you very much for your time and effort in improving our document. We have reviewed the manuscript following all the indications and suggestions of all 3 reviewers. It helps us to improve the manuscript and deliver a more comprehensive and impactful study. We made the necessary revisions to address the concerns you raised. The article was also revised by a native English speaker for language corrections. We believe that the document is much better now and hope that now you find it suitable for publication. We have carefully addressed all the comments and suggestions that the reviewer has raised. Please find below our point-by-point response to the reviewer’s comments/ suggestions and the necessary changes are highlighted with turquoise color in the revised manuscript.

  1. The introduction section should be revised to make it more concise. In this form, it contains a lot of information irrelevant to the current study. Also, the authors are advised to highlight the novelty of their work at the end of this section.

Response: Thank you for your valuable feedback. We have carefully revised the introduction section to make it more concise and focused on the current study's context. We have removed irrelevant information and ensured that the introduction now presents a clear and concise background of the research. Additionally, in the revised introduction, we have emphasized the novelty of our work at the end of the section. Please see this revised version of the manuscript.

  1. Line 171 – figure 1 should be revised as this form does not present any relevant information, it is not a scheme – only 3 glasses.

Response: Thanks, the figure 1 and 2 are merged and revised as per the valuable suggestions provided by the reviewer.

  1. Line 186 – figure 1 and Figure 2 should be joined in one figure depicting the green synthesis. Also, the synthesis steps should be presented in a more scientific way.

Response: Thanks for your suggestion. Figured 1 and 2 has been combined. Figure provided in revised manuscript is more comprehensive scientifically.

  1. Line 196 – the method used for the antifungal assay should be seriously revised as in this form is totally unclear. What is food poison method? The strain was isolated by the authors? If so, how it was identified? Where is it deposited? Why did the authors not use a standard strain for the assay? The concentration of the spores used for inoculation must be presented The PDA was supplemented with plant extract, AgNO3, and AgNPs? What kind of controls were used? How were the readings performed?

Response: The authors are thankful for valuable suggestion. The protocols for antifungal bioassay have been revised and rewritten (highlighted) addressing the comments to justify the deficiencies in the previous version. Please see lines 201-221

  1. Line 275 – Please describe in the M&M section the method for MIC determination if used in the experiments.

Response: It was a mistake. And have been corrected in the revised version of MS.

  1. Line 279 – Please explain in the M&M section how the authors calculated the inhibitory index.

Response: It was mycelial growth inhibition (%). The suggested corrections have been incorporated in the text of revised MS. Please see lines 219-221

  1. Line 285 – What is minimum activity index? Please describe how it was calculated in the M&M section. I have serious doubts that the pictures presented in Figure 8 were taken after only 24 hours of incubation because the Fusarium colonies are too big.

Response: It was F. oxysporum mycelial growth inhibition (%). The suggested corrections have been incorporated in the M&M section of revised MS. Figure 8 were taken after 96 hours of incubation.

  1. Line 290 – a reference to sustain the author’s affirmation is need it here.

Response: These lines have been shifted to discussion section and reference is provided there.

  1. Line 294 – it is not clear from the manuscript if 10 ppm to 20 ppm concentrations are for AgNPs or for A. fatua aqueous extract.

Response: The results have been thoroughly revised and rewritten to address the confusing statements and better understanding.

  1. Line 330 – what control? With deionized water or A. fatua aqueous extract?

Response: The control treatment (deionized water) was carried out without any contaminant in the PDA media. It is rectified in the text.

  1. Line 335 – Figure 8 needs a serious revision – please use only clear pictures – in this form is not clear. How was the growth assessed for B and C? What is the difference between C and G?

Response: The pictures in figure 7 (now) have been rectified with clear picture. Average diameters (mm) were calculated for all the pictures.

  1. Line 338 – the entire paragraph should be revised – in this form is not clear.

Response: The results have been thoroughly revised and rewritten to address the confusing statements and better understanding.

  1. Line 343 – Figure 9 – where are the bars for control?

Response: As, the mycelial growth inhibition (%) in control treatments was found to be zero. These bars are not visible in figures. It was calculated by the formula given by

  1. Alothman, M.; and Abd-Ei-Aziz Abeer, R. M.. Effect of green synthesis silver nanoparticles from five fruits peel on protein capped and anti-fungal properties. Int. J. Adv. Res. Biol. Sci. 2019, 6, 156–165.
  2. Line 363 – if the inhibitory index (%) is based on colony diameter measurements, Table 1 and Figures 10-12 are redundant. Also, for figures 1--12 is not clear if the concentrations of 5-40 ppm are for A. fatua extracts, AgNo3, or AgNPs.

Response: We appreciate the reviewer's observation regarding potential redundancy between Table 1 and Figures 10-12. We understand the importance of avoiding unnecessary repetition in presenting data. Our aim is to enhance the clarity, accuracy, and coherence of the content to ensure that the findings are presented in the most effective and comprehensible manner. We have explored that literature and found the same way to express such findings in many studies. Regarding the concentrations, we have clearly indicated throughout the text whether these concentrations correspond to A. fatua extracts, AgNO3, or AgNPs in the figure captions or labels.

  1. Line 447 – Please provide experimental proof. Also, please avoid speculations.

Response: This part of discussion is rewritten with scientific bases from literature. The results are justified and supported from previous studies present in the literature.

  1. The entire 4.6 Antifungal activity of green synthesized AgNPs section needs to be rewritten. In the current form is not clear, misleading, full of speculations, and useless information. The authors are encouraged to discuss their own results.

Response: The discussion is improved and rewritten by providing the supports from the various studies conducted in the recent past about the antifungal activities and mechanisms described by different authors.

Reviewer 3 Report

Considering my expertise (General microbiology), the current revision will cover only the antifungal tests.

The main goal of the paper was to investigate the potential of Avena fatua as a green source of AgNps with antifungal activity. However, the idea of using plants for the synthesis of silver nanoparticles with antimicrobial properties is not new and from this perspective the main drawback of the manuscript is the lack of novelty. In addition, in this form the manuscript is unacceptable for publication as it does not comply with minimum scientific standards of writing a paper.

The introduction section should be revised to make it more concise. In this form it contains a lot of information irrelevant to the current study. Also, the authors are advised to highlight the novelty of their work at the end of this section.

Line 171 – figure 1 should be revised as this form does not present any relevant information, it is not a scheme – only 3 glasses.

Line 186 – figure 1 and figure 2 should be joined in one figure depicting the green synthesis. Also, the synthesis steps should be presented in a more scientific way.

 Line 196 – the method used for the antifungal assay should be seriously revised as in this form is totally unclear. What is food poison method? The strain was isolated by the authors? If so, how it was identified? Where is it deposited? Why did the authors not use a standard strain for the assay? The concentration of the spores used for inoculation must be presented The PDA was supplemented with plant extract, AgNO3 and AgNPs? What kind of controls were used? How were the readings performed?

Line 275 – Please describe in the M&M section the method for MIC determination if used in the experiments.

Line 279 – Please explain in the M&M section how the authors calculated the inhibitory index.

Line 285 – what is minimum activity index? Please describe how it was calculated in the M&M section. I have serious doubts that the pictures presented in Figure 8 were taken after only 24 hours of incubation because the Fusarium colonies are too big.

Line 290 – a reference to sustain the author’s affirmation is need it here.

Line 294 – it is not clear from the manuscript if 10 ppm to 20 ppm concentration are for AgNPs or for A. fatua aqueous extract.

Line 330 – what control? With deionized water or A. fatua aqueous extract?

Line 335 – Figure 8 needs a serious revision – please use only clear pictures – in this form is not clear. How was the growth assessed for B and C? What is the difference between C and G?

Line 338 – the entire paragraph should be revised – in this form is not clear.

Line 343 – Figure 9 – where are the bars for control?

Line 363 – if the inhibitory index (%) is based on colony diameter measurements, Table 1 and Figures 10-12 are redundant. Also, for the figures 1--12 is not clear if the concentrations of 5-40 ppm are for A fatua extracts, AgNo3 or AgNPs.

Line 447 – please provide experimental proofs. Also, please avoid speculations.

The entire 4.6 Antifungal activity of green synthesized AgNPs section needs to be re-written. In the current form is not clear, misleading, full of speculations and useless information. The authors are encouraged to discuss their own results.

Author Response

Dear Reviewer, #3,

Many thanks for pointing out the deficiencies, providing useful suggestions and the opportunity to revise our manuscript. We have completely revised the whole manuscript and considering the given suggestions accepting all of them. The said changes have been incorporated in relevant places highlighted with green color.

Comments and Suggestions for Authors

The work is well written, the methodology is correct and also the results and discussion. However, the most important aspect to consider this paper acceptable for publication is the novelty. Many works have reported the biogenic synthesis of metals nanoparticles and their antifungal properties it is no a novelty for the readers. Then, the authors need to highlight the novelty of this manuscript or add new and more relevant information.

Response: We would like to extend our heartfelt gratitude to Reviewer for his invaluable contribution to the review process of our submitted manuscript and appreciation for our work. We have carefully addressed all the comments and suggestions that reviewer has raised. To the best of our knowledge, A. fatua was never employed for the synthesis of any type of nanomaterials and never tested for antimicrobial potential. This is the first time when A. fatua was explored for its antimicrobial potential against F. oxysporum through greener synthesis AgNPs.

  1. On the other hand, why the authors used AgNO3 as control? This material is not frequently used as antifungal. Why the authors not used a common antifungal?

Response: As we have prepared the green AgNPs by using A. fatua extracts. The AgNPs were coated with A. fatua extracts which served as a reducing agent during the nanoparticle’s synthesis. That’s why we have to use A. fatua extracts and AgNO3 as treatments in addition to control (dd water) for a better comparison of antifungal activity of AgNPs with A. fatua extracts and AgNO3. Common antifungal was not necessary to be used as we were comparing the green synthesized AgNPs with AgNO3 only.

  1. In Table 1, please only 1 decimal after point.

Response: Done, thank you.

Round 2

Reviewer 2 Report

The authors have been enhanced the manuscript according to the suggestion of the reviewers.

Author Response

Dear Reviewer,

thank you for your last comments, as you have seen, we included all of them and we would like to thank you and all the reviewers in the last versions we uploaded. 

Thank you again

Sincerely 

Barbara Manachini on the behalf of all authors

Reviewer 3 Report

The improvement of the manuscript following the reviewers’ suggestions is visible. However, more corrections are needed to achieve a better quality for publications.

Line 148 – please revise the caption title of Figure 1 to explain better the scheme (the figure does not represent the brownish red color of synthesized AgNPs)

Line 163 – please revise Antifungal Bioassay, especially the lines 168-174. Several questions addressed before remained without answer e.g. the use of a standard strain for the assay, the spores concentration used for inoculation; also, please provide institution and country for FMB-CC-UAF.

Line 338 – the authors are encouraged to revise Figure 7 – in this form it cannot be considered paper quality.

Line 348 – please explain the significance of the different letters used in Table 1.

Line 289 – revise the two sentences – maybe merge in only one sentence. Same for line 304.

Line 305 – avoid immense in the sentence.

Line 343 – where is the control depicted in the figure? We have it in the legend but not in the graph. Same for all graphs.

Line 382 – Please revise as it is not clear which is the subject of the sentence - green synthesis or silver nanoparticles.

Line 460 – Please provide scientific data to sustain the synergic effect. If not, please remove it from the manuscript because is a simple speculation.

Line 468 – please revise the sentence as it is too long.

Line 479 – why literature comparisons with Aspergillus and not with Fusarium?

The authors are encouraged once again to revise the discussion section and to make it clearer and more concise, with respect to their own findings and literature data.

Author Response

Dear reviewer,

thank you for your time and your useful suggestions that once again improved our paper. We accept all your comments and indications. 

Please find below our response in detail.

Thank you again

REVIEWER 3 (R3) The improvement of the manuscript following the reviewers’ suggestions is visible. However, more corrections are needed to achieve a better quality for publications.

Thank you very much for your time and effort in improving our document.
Comments and Suggestions for Authors were all included. 

R3. Line 148 – please revise the caption title of Figure 1 to explain better the scheme (the figure does not represent the brownish red color of synthesized AgNPs)
Response: Thank you very much for suggestion. Figure 1 caption has been revised. Please see this revised version.

R3. Line 163 – please revise Antifungal Bioassay, especially the lines 168-174. Several questions addressed before remained without answer e.g., the use of a standard strain for the assay, the spore concentration used for inoculation; also, please provide institution and country for FMB-CC-UAF.Response: Thank you so much for your insightful comment. Suggestion regarding strain has been incorporated. Institution and country name has been added. Furthermore, we didn’t use any spore concentration for antifungal activity. A mycelial disc from the active growing margins of culture was cut with the help of a Sterile cork borer and was subsequently placed in the center of the treated plates. Here are a few more references to follow describing this standard protocol.
https://www.mdpi.com/2079-4991/13/7/1274

R3. Line 289 – revise the two sentences – maybe merge in only one sentence. Same for line 304.
Response: Thanks, it is a valuable suggestion, and these lines are revised and merged. Please see this revised version.

R3. Line 305 – avoid immense in the sentence.
Response: Done, thank you so much.

R3 Line 338 – the authors are encouraged to revise Figure 7 – in this form it cannot be considered paper quality.
Response: Thank you for your valuable feedback. We have carefully revised figure 7 now.

R3. Line 348 – please explain the significance of the different letters used in Table 1.
Response: Thank you for the suggestion. The suggested explanation has been provided in this revised version. Please see Table 1.

R3. Line 343 – where is the control depicted in the figure? We have it in the legend but not in the graph. Same for all graphs.
Response: The mycelial growth inhibition (%) in control treatments was found to be zero which is why these bars are not visible in the graphs. Thank you for understanding.

R3. Line 382 – Please revise as it is not clear which is the subject of the sentence - green synthesis or silver nanoparticles.
Response: The authors are thankful for the valuable suggestion. These lines have been revised as per the suggestion of the worthy reviewer.

R3. Line 460 – Please provide scientific data to sustain the synergic effect. If not, please remove it from the manuscript because is a simple speculation.
Response: The authors are thankful for valuable corrections and unnecessary words are removed from these lines.

R3. Line 468 – please revise the sentence as it is too long.
Response: The authors are thankful for the valuable correction. These lines have been rectified.

R3. Line 479 – why literature comparisons with Aspergillus and not with Fusarium?
Response: The authors are thankful for the valuable suggestion. The indicated study is removed and findings are discussed with reference to antifungal activities of AgNPs. 

R3. The authors are encouraged once again to revise the discussion section and to make it clearer and more concise, with respect to their own findings and literature data.
Response: The authors are thankful for valuable suggestions and findings are discussed with reference to our own results for a better preview of the findings.

Round 3

Reviewer 3 Report

Line 177 – please remove the next sentence: The media was poured into Petri dishes on the spirit lamp and was shaken gently to mix the solution with the media thoroughly. – it is only a simple repetition of previous information presented at line 175.

Line 341 – please remove the control from Figure 8, as well as for all following graphs – it is 0 and makes no sense.

Author Response

Dear Reviewer,

Many thanks for pointing out the suggestions, and providing valuable improvements for a better understanding of our manuscript. We have completely revised the manuscript while removing the pointed flaws and considering the given directions.

Line 177 – Please remove the next sentence: The media was poured into Petri dishes on the spirit lamp and was shaken gently to mix the solution with the media thoroughly. – it is only a simple repetition of previous information presented in line 175.

Response: Thank you for indicating the repetition. These lines have been removed.

Line 341 – please remove the control from Figure 8, as well as for all following graphs – it is 0 and makes no sense.

Response: Thanks for your suggestion. The suggested Figures are revised in the manuscript to make a clearer understanding.